# CROSS-VIEW TRAINING FOR SEMI-SUPERVISED LEARNING

## ABSTRACT

We present Cross-View Training (CVT), a simple but effective method for deep semi-supervised learning. On labeled examples, the model is trained with standard cross-entropy loss. On an unlabeled example, the model first performs inference (acting as a "teacher") to produce soft targets. The model then learns from these soft targets (acting as a "student"). We deviate from prior work by adding multiple auxiliary student prediction layers to the model. The input to each auxiliary student layer is a sub-network of the full model that has a restricted view of the input (e.g., only seeing one region of an image). The students can learn from the teacher (the full model) because the teacher sees more of each example. Concurrently, the students improve the quality of the representations used by the teacher as they learn to make predictions with limited data. When combined with Virtual Adversarial Training, CVT improves upon the current state-of-the-art on semi-supervised CIFAR-10 and semi-supervised SVHN. We also apply CVT to train models on five natural language processing tasks using hundreds of millions of sentences of unlabeled data. On all tasks CVT substantially outperforms supervised learning alone, resulting in models that improve upon or are competitive with the current state-of-the-art.

## 1 INTRODUCTION

Deep learning classifiers work best when trained on large amounts of labeled data. However, acquiring labels can be costly, motivating the need for effective semi-supervised learning techniques that leverage unlabeled examples during training. Many semi-supervised learning algorithms rely on some form of self-labeling. In these approaches, the model acts as both a "teacher" that makes predictions about unlabeled examples and a "student" that is trained on the predictions. As the teacher and the student have the same parameters, these methods require an additional mechanism for the student to benefit from the teacher's outputs.

One approach that has enjoyed recent success is adding noise to the student's input (Bachman et al., 2014; Sajjadi et al., 2016). The loss between the teacher and the student becomes a consistency cost that penalizes the difference between the model's predictions with and without noise added to the example. This trains the model to give consistent predictions to nearby data points, encouraging smoothness in the model's output distribution with respect to the input. In order for the student to learn effectively from the teacher, there needs to be a sufficient difference between the two. However, simply increasing the amount of noise can result in unrealistic data points sent to the student. Furthermore, adding continuous noise to the input makes less sense when the input consists of discrete tokens, such in natural language processing.

We address these issues with a new method we call Cross-View Training (CVT). Instead of only training the full model as a student, CVT adds auxiliary softmax layers to the model and also trains them as students. The input to each student layer is a sub-network of the full model that sees a restricted view of the input example (e.g., only seeing part of an image), an idea reminiscent of co-training (Blum & Mitchell, 1998). The full model is still used as the teacher. Unlike when using a large amount of input noise, CVT does not unrealistically alter examples during training. However, the student layers can still learn from the teacher because the teacher has a better, unrestricted view of the input. Meanwhile, the student layers improve the model's representations (and therefore the

teacher) as they learn to make accurate predictions with a limited view of the input. Our method can be easily combined with adding noise to the students, but works well even when no noise is added.

We propose variants of our method for Convolutional Neural Network (CNN) image classifiers, Bidirectional Long Short-Term Memory (BiLSTM) sequence taggers, and graph-based dependency parsers. For CNNs, each auxiliary softmax layer sees a region of the input image. For sequence taggers and dependency parsers, the auxiliary layers see the input sequence with some context removed. For example, one auxiliary layer is trained to make predictions without seeing any tokens to the right of the current one.

We first evaluate Cross-View Training on semi-supervised CIFAR-10 and semi-supervised SVHN. When combined with Virtual Adversarial Training (Miyato et al., 2017b), CVT improves upon the current state-of-the-art on both datasets. We also train semi-supervised models on five tasks from natural language processing: English dependency parsing, combinatory categorical grammar supertagging, named entity recognition, text chunking, and part-of-speech tagging. We use the 1 billion word language modeling benchmark (Chelba et al., 2014) as a source of unlabeled data. CVT works substantially better than purely supervised training, resulting in models that improve upon or are competitive with the current state-of-the-art on every task. We consider these results particularly important because many recently proposed semi-supervised learning methods work best on continuous inputs and have only been evaluated on vision tasks (Bachman et al., 2014; Sajjadi et al., 2016; Laine & Aila, 2017; Tarvainen & Valpola, 2017). In contrast, CVT can handle discrete inputs such as language very effectively.

## 2 RELATED WORK

Semi-supervised learning in general has been widely studied (Chapelle et al., 2006). Early approaches to deep semi-supervised learning pre-train neural models on unlabeled data, which has been successful for applications in computer vision (Jarrett et al., 2009; LeCun et al., 2010) and natural language processing (Dai & Le, 2015; Ramachandran et al., 2017). More recent work incorporates generative models based on autoencoders (Kingma et al., 2014; Rasmus et al., 2015) or Generative Adversarial Networks (Springenberg, 2015; Salimans et al., 2016) into the training.

**Self-Training.** One of the earliest approaches to semi-supervised learning is self-training (Scudder, 1965; Fralick, 1967). Initially, a classifier is trained on labeled data only. In each subsequent round of training, the classifier, acting as a "teacher," labels some of the unlabeled data and adds it to the training set. Then, acting as a "student," it is retrained on the new training set. The new examples added each round act as noisy "pseudo labels" (Lee, 2013) that the model can learn from. Many recent approaches train the student with soft targets from the teacher's output distribution rather than a hard label, making the procedure more akin to knowledge distillation (Hinton et al., 2015).

**Consistency Training and Distributional Smoothing.** Recent works add noise to the student's input (Bachman et al., 2014; Sajjadi et al., 2016). This trains the model to give consistent predictions to nearby data points, encouraging distributional smoothness in the model. Inspired by the success of adversarial training (Goodfellow et al., 2015), Miyato et al. (2016) extend this idea by adversarially selecting the perturbation to the input. Other approaches focus on improving the targets provided by the teacher by tracking an exponential moving average of its predictions (Laine & Aila, 2017) or its weights (Tarvainen & Valpola, 2017). Our method is complimentary to these previous approaches, and can be combined with them effectively.

**Co-Training.** Co-Training (Blum & Mitchell, 1998; Nigam & Ghani, 2000) trains two models with disjoint views of the input. On unlabeled data, each one acts as a "teacher" for the other model. In contrast, our approach trains a single unified model where auxiliary prediction layers see different, but not necessarily independent views of the input.

**Auxiliary Prediction Layers.** Another way of leveraging unlabeled data is through the addition of auxiliary "self-supervised" losses. These approaches train auxiliary prediction layers on tasks where performance can be measured without human-provided labels. Previous work has jointly trained image classifiers with tasks like relative position and colorization (Doersch & Zisserman, 2017), sequence taggers with language modeling (Rei, 2017), and reinforcement learning agents

Figure 1: An overview of Cross-View Training. The model is trained with standard supervised learning on labeled examples. On unlabeled examples, auxiliary softmax layers with different views of the input are trained to agree with the primary softmax layer. Although the model takes on different roles (i.e., as the teacher or the student), only one set of parameters is trained.

with predicting changes in the environment (Jaderberg et al., 2017). Unlike these approaches, our auxiliary losses are based on self-labeling, not labels deterministically constructed from the input.

**Data Augmentation.** Data augmentation, such as random translations or crops of input images, bears some similarity to our method in that it also exposes the model to different views of input examples. Data augmentation has become a common practice for both supervised and semi-supervised training of image classifiers (Simard et al., 2003; Krizhevsky et al., 2012).

## 3 CROSS-VIEW TRAINING

We first provide a general description of Cross-View Training. We then present specific constructions for auxiliary prediction layers that work well for image classification, sequence tagging, and dependency parsing.

### 3.1 METHOD

We use $\mathcal{D}_l = \{(x_1, y_1), (x_2, y_2), ..., (x_N, y_N)\}$ to represent a labeled dataset and $\mathcal{D}_{ul} = \{x_1, x_2, ..., x_M\}$ to represent an unlabeled dataset. We use $p_\theta(y|x_i)$ to denote the output distribution over classes produced by a model with parameters $\theta$ on input $x_i$. Our approach uses a standard cross-entropy loss over the labeled data:

$$\mathcal{L}_{\text{sup}}(\theta) = \frac{1}{|\mathcal{D}_l|} \sum_{x_i, y_i \in \mathcal{D}_l} CE(y_i, p_\theta(y|x_i))$$

On unlabeled data, a popular approach is to add a consistency cost encouraging distributional smoothness in the model. First, the model produces soft targets for the current example: $\hat{y}_i = p_\theta(y|x_i)$. The model is then trained to minimize the consistency cost

$$\mathcal{L}_{\text{consistency}}(\theta) = \frac{1}{|\mathcal{D}_{ul}|} \sum_{x_i \in \mathcal{D}_{ul}} \mathbb{E}_\eta \left[ D(\hat{y}_i, p_\theta(y|x_i + \eta)) \right]$$

where $D$ is a distance function (we use KL divergence) and $\eta$ is a perturbation to the input that can be chosen randomly or adversarially. As is common in prior work, we hold the teacher's prediction $\hat{y}_i$ fixed during training (i.e., we don't back-propagate through it) so the student learns to imitate the teacher, but not vice versa.

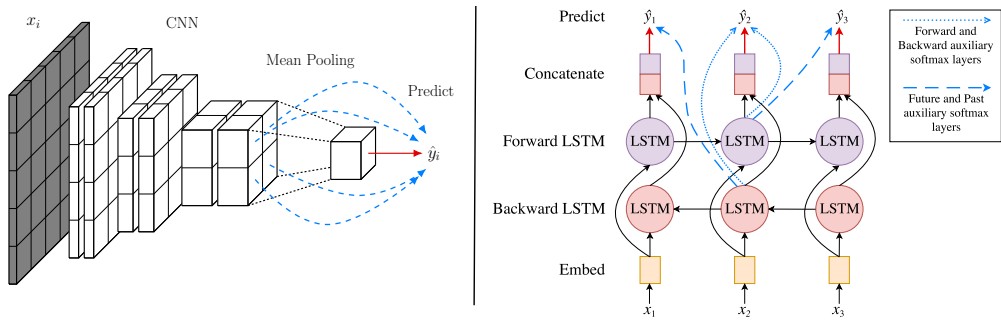

Figure 2: Softmax layers for image classifiers and sequence taggers. Solid red arrows represent primary softmax layers and dashed blue arrows represent auxiliary ones. To simplify the diagram, the CNN only produces four feature vectors, the BiLSTM only has a single layer, and we only show auxiliary layers for the BiLSTM's second time step. For CNNs, auxiliary layers take a single feature vector as input. For BiLSTMs, auxiliary layers are attached to the forward and backward LSTMs. Our dependency parsing models use auxiliary layers analogous to the "forward" and "backward" sequence tagging ones.

Cross-View Training adds $k$ additional prediction layers $p_\theta^1, ..., p_\theta^k$ to the model. Each layer $p_\theta^j$ takes as input an intermediate representation $h_j(x_i)$ produced by the model. It outputs a distribution over labels, usually with a softmax layer (an affine transformation followed by a softmax activation function) applied to this representation: $p_\theta^j(y|x_i) = \texttt{SML}(h_j(x_i)) = \text{softmax}(W_j h_j(x_i) + b_j)$. At test time, only the main prediction layer $p_\theta$ is used. Each $h_j$ is chosen such that it only uses a part of each input $x_i$; the particular choice can depend on the task and model architecture. We propose variants for CNN image classifiers, BiLSTM sequence taggers, and graph-based dependency parsers in sections 3.2, 3.3, and 3.4. We add the distances between the output distributions of the teacher and auxiliary students to the consistency loss, resulting in a cross-view consistency (CVC) loss:

$$\mathcal{L}_{\text{CVC}}(\theta) = \frac{1}{|\mathcal{D}_{ul}|} \sum_{x_i \in \mathcal{D}_{ul}} \mathbb{E}_\eta \left[ D(\hat{y}_i, p_\theta(y|x_i + \eta)) + \frac{\lambda_1}{k} \sum_{j=1}^{k} D(\hat{y}_i, p_\theta^j(y|x_i + \eta)) \right]$$

We combine the supervised and CVC losses into the total loss, $\mathcal{L} = \mathcal{L}_{\text{sup}} + \lambda_2 \mathcal{L}_{\text{CVC}}$, and minimize it with stochastic gradient descent. At each step, $\mathcal{L}_{\text{sup}}$ is computed over a minibatch of labeled examples and $\mathcal{L}_{\text{CVC}}$ is computed over a minibatch of unlabeled examples. $\lambda_1$ and $\lambda_2$ are hyperparameters controlling the strength of the auxiliary prediction layers and the strength of the unsupervised loss. For all experiments we set $\lambda_1 = k$ and $\lambda_2 = 1$ unless indicated otherwise. See Figure 1 for an illustration of the training procedure.

Although adding noise or an adversarial perturbation to the input generally improves results, $\mathcal{L}_{\text{CVC}}$ can be trained without this enhancement (i.e., setting $\eta = 0$). In this case, the first term inside the expectation disappears (the student will exactly match the teacher, so the distance is zero). In contrast, $\mathcal{L}_{\text{consistency}}$ requires a nonzero $\eta$ to make the student and teacher output different distributions.

In most neural networks, a few additional softmax layers is computationally cheap compared to the portion of the model building up representations (such as a CNN or RNN). Therefore our method contributes little overhead to training time over consistency training. CVT does not change inference time because the auxiliary layers are only used during training.

## 3.2 IMAGE RECOGNITION MODELS

Our image recognition models are based on Convolutional Neural Networks, which produce a set of features $H(x_i) \in \mathbb{R}^{n \times n \times d}$ from an image $x_i$. The first two dimensions of $H$ index into the spatial coordinates of feature vectors and $d$ is the size of the feature vectors. For shallower CNNs, a particular feature vector corresponds to a region of the input image. For example, $H_{0,0}$ would

be a $d$-dimensional vector of features extracted from the upper left corner. For deeper CNNs, a particular feature vector would be extracted from the whole image, but still only use a "region" of the representations from an earlier layer. The CNNs in our experiment are all in the first category.

The primary prediction layer for our CNNs take as input the mean of $H$ over the first two dimensions, which results in a $d$-dimensional vector that is fed into a softmax layer:
$p_\theta(y|x_i) = \texttt{SML}(\texttt{global\_average\_pool}(H))$.

We add $n^2$ auxiliary softmax layers to the top of the CNN. The $j$th layer takes a single feature vector as input, as shown in the left of Figure 2: $p_\theta^j(y|x_i) = \texttt{SML}(H_{\lfloor j/n \rfloor, j \bmod n})$. We also experimented with adding auxiliary softmaxes to the outputs of earlier layers in the CNN, but found this did not improve performance.

## 3.3 SEQUENCE TAGGING MODELS

In sequence tagging, each example $(x_i, y_i)$ consists of $T$ tokens $x_i^1, ..., x_i^T$ and $T$ corresponding labels $y_i^1, ..., y_i^T$. We assume an $L$-layer bidirectional RNN sequence tagging model, which has become standard for many sequence tagging tasks (Graves & Schmidhuber, 2005; Graves et al., 2013). Each layer runs an RNN such as an LSTM (Hochreiter & Schmidhuber, 1997) in the forward direction (taking $x_i^t$ as input at each step $t$) and the backward direction (taking $x_i^{T-t+1}$ as input at each step) and concatenates the results. A softmax layer on top of the outputs of the last BiRNN layer, $h_L(x_i) = [h_L^1(x_i), ..., h_L^T(x_i)]$, makes the predictions: $p_\theta(y^t|x_i) = \texttt{SML}(h_L^t(x_i))$.

The auxiliary softmax layers take $\overrightarrow{h}_1(x_i)$ and $\overleftarrow{h}_1(x_i)$, the outputs of the forward and backward RNNs in the first BiRNN layer, as inputs. We add the following four softmax layers to the model (see the right of Figure 2):

$$p_\theta^{\text{fwd}}(y^t|x_i) = \texttt{SML}(\overrightarrow{h}_1^t(x_i)) \qquad p_\theta^{\text{bwd}}(y^t|x_i) = \texttt{SML}(\overleftarrow{h}_1^t(x_i))$$
$$p_\theta^{\text{future}}(y^t|x_i) = \texttt{SML}(\overrightarrow{h}_1^{t-1}(x_i)) \qquad p_\theta^{\text{past}}(y^t|x_i) = \texttt{SML}(\overleftarrow{h}_1^{t+1}(x_i))$$

The "forward" and "backward" prediction layers use the RNN's current output to predict the current token. The "future" and "past" layers use the RNN's previous output (or, equivalently, they predict the label for the next token). The forward layer makes each prediction without seeing the right context of the current token. The future layer makes each prediction without the right context or the current token itself. Therefore it works like a neural language model that, instead of predicting which token comes next in the sequence, predicts which class of token comes next in the sequence.

## 3.4 DEPENDENCY PARSING MODELS

In a dependency parse, words in a sentence are treated as nodes in a graph. Typed directed edges connect the words, forming a tree structure describing the syntactic structure of the sentence. In particular, each word $x_i^t$ in a sentence $x_i = x_i^1, ..., x_i^T$ receives exactly one in-going edge $(u, t, r)$ going from word $x_i^u$ (called the "head") to it (the "dependent") of type $r$ the (the "relation"). Therefore a dependency parsing example consists of $T$ tokens $x_i = x_i^1, ..., x_i^T$ and $T$ corresponding labels $y_i = y_i^1, ..., y_i^T$ where each label $y_i^t$ represents the in-going edge to word $x_i^t$: $y_i^t = (u, t, r)$. To give a specific example, in the sentence "The small dog barked", the correct label for "small" would be the edge ("dog", "small", $\texttt{adjectival-modifier}$).

We use a neural graph-based dependency parser similar to the one from Dozat & Manning (2017). It first runs a BiRNN encoder over the sentence as described in section 3.3, producing a sequence of outputs $h_L(x_i) = [h_L^1(x_i), ..., h_L^T(x_i)]$. Each output $h_L^t(x_i)$ is passed through two separate multi-layer perceptrons, one producing a representation for $x_i^t$ as a head word and one producing a representation for it as a dependent. A bilinear classifier applied to these representations produces a score for each candidate edge. Lastly, these scores are passed through a softmax layer to produce probabilities. Mathematically, the probability of an edge is given as $p_\theta((u, t, r)|x_i) \propto e^{s(h_L^u(x_i), h_L^t(x_i), r)}$. Where $s$ is the scoring function $s(z_1, z_2, r) = \texttt{MLP}_{\text{head}}(z_1)(W_r + W)\texttt{MLP}_{\text{dep}}(z_2)$. The bilinear classifier uses a weight matrix $W_r$ specific to the candidate relation as well as a weight matrix $W$ shared across all relations.

We add four auxiliary prediction layers to our model for cross-view training:

$$p_\theta^{\text{fwd-fwd}}((u,t,r)|x_i) \propto e^{s(\overrightarrow{h}_1^u(x_i), \overrightarrow{h}_1^t(x_i),r)} \qquad p_\theta^{\text{fwd-bwd}}((u,t,r)|x_i) \propto e^{s(\overrightarrow{h}_1^u(x_i), \overleftarrow{h}_1^t(x_i),r)}$$

$$p_\theta^{\text{bwd-fwd}}((u,t,r)|x_i) \propto e^{s(\overleftarrow{h}_1^u(x_i), \overrightarrow{h}_1^t(x_i),r)} \qquad p_\theta^{\text{bwd-bwd}}((u,t,r)|x_i) \propto e^{s(\overleftarrow{h}_1^u(x_i), \overleftarrow{h}_1^t(x_i),r)}$$

Each auxiliary layer has some missing context (not seeing either the preceding or following words) for the candidate head and candidate dependent. All the parameters for the scoring function of each auxiliary prediction layer are layer-specific.

## 4 EXPERIMENTS

To validate our approach, we evaluate Cross-View Training on two semi-supervised learning benchmarks. These discard most of the labels from standard image image recognition datasets to artificially make them semi-supervised. As a sterner test of our approach, we also apply CVT to five tasks from Natural Language Processing (NLP) using hundreds of millions of unlabeled sentences for semi-supervised learning.

### 4.1 IMAGE RECOGNITION

**Data.** We experiment on two semi-supervised image recognition benchmarks. These are constructed from the CIFAR-10 (Krizhevsky & Hinton, 2009) and Street View House Numbers (SVHN) (Netzer et al., 2011) datasets. Following previous work, we make the datasets semi-supervised by only using the provided labels for a subset of the examples in the training set; the rest are treated as unlabeled examples.

**Model.** We use the convolutional neural network from Miyato et al. (2017b), adapting their Tensor-Flow implementation[1]. Their model, based on Springenberg et al. (2014), contains 9 convolutional layers and 2 max pooling layers. See Appendix D of Miyato et al.'s paper for details.

We add 36 auxiliary softmax layers to the $6 \times 6$ collection of feature vectors produced by the CNN. Each auxiliary layer sees a patch of the image ranging in size from $21 \times 21$ pixels (the corner) to $29 \times 29$ pixels (the center) of the $32 \times 32$ pixel images. We optimize $\mathcal{L}$ with $\lambda_1 = 1$ and each minibatch consisting of 32 labeled and 128 unlabeled examples.

Miyato et al. use Virtual Adversarial Training (VAT), minimizing $\mathcal{L}_{\text{consistency}}$ with the input perturbation $\eta$ chosen adversarially. We train our cross-view models (which instead use $\mathcal{L}_{\text{CVC}}$) both with and without this adversarial noise. We report results with and without using data augmentation (random translations for SVHN and random translations and horizontal flipping for CIFAR-10) in Table 1.

**Results.** CVT works well as semi-supervised learning method without any noise being added to the student. When random noise is added, it performs close to VAT (the standard-deviation-based confidence intervals intersect) while training much faster (requiring only one backwards pass for each training minibatch, while VAT requires an additional one to compute the adversarial perturbation). Our method can easily be combined with VAT, resulting in further improvements and state-of-the-art results. The benefit of CVT is less when data augmentation is applied, perhaps because random translations of the input expose the model to different "views" in a similar manner as with CVT. We believe the gains on SVHN are smaller than CIFAR-10 because the digits in SVHN occur in the center of the image, so the auxiliary softmaxes seeing the sides and corner do not learn as effectively. We also note that incorporating auxiliary softmax layers into the supervised loss $\mathcal{L}_{\text{sup}}$ does not improve results (see Appendix C). This indicates that the benefit of CVT comes from the improved self-training mechanism, not the additional losses regularizing the model.

**Model Analysis.** To understand why CVT produces better results, we compare the behavior of the VAT and CVT (with adversarial noise) models trained on CIFAR-10. First, we record the average value of each feature vector produced by the CNNs when they run over the test set. As shown in the left of Figure 3, the CVT model has higher activation strengths for the feature vectors corresponding to the edges of the image. We hypothesize that the VAT model fits to the data while primarily using

---

[1]https://github.com/takerum/vat_tf

| Method | SVHN | SVHN+ | CIFAR-10 | CIFAR-10+ |
|---|---|---|---|---|
| | 1000 labels | | 4000 labels | |
| GAN[a] | – | $8.11 \pm 1.3$ | – | $18.63 \pm 2.32$ |
| Stochastic Transformations[b] | – | – | – | $11.29 \pm 0.24$ |
| Π model[c] | $5.43 \pm 0.25$ | $4.82 \pm 0.17$ | $16.55 \pm 0.29$ | $12.36 \pm 0.31$ |
| Temporal Ensemble[c] | – | $4.42 \pm 0.16$ | – | $12.16 \pm 0.24$ |
| Mean Teacher[d] | – | $\mathbf{3.95 \pm 0.19}$ | – | $12.31 \pm 0.28$ |
| Complement GAN[e] | $4.25 \pm 0.03$ | – | $14.41 \pm 0.30$ | – |
| VAT[f] | 4.28 | **3.86** | 13.15 | 10.55 |
| Supervised | $10.68 \pm 0.51$ | $10.10 \pm 0.48$ | $23.61 \pm 0.60$ | $19.61 \pm 0.56$ |
| VAT* | $4.11 \pm 0.13$ | $\mathbf{3.83 \pm 0.16}$ | $13.29 \pm 0.33$ | $10.90 \pm 0.31$ |
| CVT, no noise | $4.48 \pm 0.09$ | $4.37 \pm 0.12$ | $14.63 \pm 0.20$ | $12.44 \pm 0.27$ |
| CVT, random noise | $4.11 \pm 0.08$ | $4.04 \pm 0.16$ | $13.80 \pm 0.30$ | $11.10 \pm 0.26$ |
| CVT, adversarial noise | $\mathbf{3.79 \pm 0.08}$ | $\mathbf{3.70 \pm 0.15}$ | $\mathbf{12.01 \pm 0.11}$ | $\mathbf{10.11 \pm 0.15}$ |

[a]Salimans et al. (2016) [b]Sajjadi et al. (2016) [c]Laine & Aila (2017) [d]Tarvainen & Valpola (2017)
[e]Dai et al. (2017) [f]Miyato et al. (2017b)
*We found Miyato et al.'s implementation produces slightly different results than the ones they report in their paper.

Table 1: Error rates on semi-supervised learning benchmarks. We report means and standard deviations from 5 runs. + after a dataset means data augmentation was applied.

Figure 3: Left: Ratio between the average activation of feature vectors from final layer of the CVT CNNs divided by the average from the VAT CNNs. Each square in a grid represents a single feature vector. Brighter means the feature vectors from the CVT model are more activated. Center, Right: Accuracy of prediction layers taking a single feature vector as input. The CVT model makes more use of the outside of the image and produces better representations for those regions

the center of the image, where the most discriminative information is contained. This results in less effective feature vectors for the outside regions. In contrast, the model with CVT must learn meaningful representations for the edge regions in order to train the corresponding auxiliary softmax layers. As these feature vectors are more useful, their magnitude become larger so they contribute more to the final representation produced by the global average pool.

To compare to discriminatory power of the feature vectors, we freeze the weights of the CNNs and add auxiliary softmax layers that are trained from scratch. We then measure the accuracies of the added layers (see the center and right of Figure 3). Unsurprisingly, the VAT model, which only learns representations that will be useful after the average pool, has much lower accuracies from individual feature vectors. The difference is particularly striking in the sides and corners, where CVT accuracies are around 50% higher (they are about 25% higher in the center). This finding further indicates that CVT is improving the model's representations, particularly for the outside parts of images.

## 4.2 NATURAL LANGUAGE PROCESSING TASKS

**Data.** Although the widely-used benchmarks in the previous section provide validation of our approach, they are small datasets that are artificially made to be semi-supervised. In this section, we show CVT is successful on well-studied tasks where semi-supervised learning is rarely applied. In particular, we train semi-supervised models on the following NLP tasks:

- **Combinatory Category Grammar (CCG) Supertagging:** Labeling words with CCG supertags: lexical categories that encode information about the predicate-argument structure of the sentence. CCG is widely used in syntactic and semantic parsing. We use data from CCGBank (Hockenmaier & Steedman, 2007) and report word-level accuracy.

- **Text Chunking**: Dividing a sentence into syntactically correlated parts (e.g., a noun phrase followed by a verb phrase). We use the CoNLLL-2000 shared task data (Tjong Kim Sang & Buchholz, 2000) and report the F1 score over predicted chunks.

- **Named Entity Recognition (NER)**: Identifying and classifying named entities (organizations, places, etc.) in a sentence. We use the CoNLL-2003 dataset (Tjong Kim Sang & De Meulder, 2003) and report entity-level F1 score.

- **Part-of-Speech (POS) Tagging**: Labeling words with their syntactic categories (e.g., determiner, adjective, etc.). We use the Wall Street Journal (WSJ) portion of the Penn Treebank (Marcus et al., 1993) and report word-level accuracy.

- **Dependency Parsing:** Inferring a tree-structure describing the syntactic structure of a sentence. We use the Penn Treebank converted to Stanford Dependencies (version 3.3.0) and report unlabeled and labeled attachment score (UAS and LAS).

We use the 1 Billion Word Language Model Benchmark (Chelba et al., 2014) as a pool of unlabeled sentences for semi-supervised learning.

**Models.** We use a CNN-BiLSTM sequence tagging model (Chiu & Nichols, 2016; Ma & Hovy, 2016). The model first represents each word as the sum of a word embedding and the output of a character-level CNN. This sequence of word representations is then fed through two BiLSTM layers and a softmax layer to produce predictions. See Appendix A for details about the model.

Our dependency parser uses the same CNN-BiLSTM encoder as our sequence tagger. As described in Section 3.4, a MLP-Bilinear classifier on top of the encoder makes the predictions. Although it is common for dependency parsers to take words and part-of-speech tags as inputs, our model only takes words as inputs. See Appendix B for details about the model.

Miyato et al. (2017a) were able to apply Virtual Adversarial Training to document classification, but we found VAT ineffective for our word-level tasks. Although we experimented with constraining the word embeddings to unit length and adding random or adversarial perturbations to them during training, it did not improve performance. This is perhaps because, unlike with RGB values in an image, words are discrete, so adding noise to their representations is less meaningful. Instead, we add dropout to the student but not the teacher.

Recent work (Rei, 2017; Liu et al., 2017) has shown that jointly training a neural language model with sequence taggers improves results. We report accuracies with and without this enhancement (training the language model on the unlabeled data). See Table 2 for sequence tagging results and Table 3 for dependency parsing results.

**Results.** CVT significantly improves over the supervised baseline on all tasks, both with and without the auxiliary language modeling objective. We report a new state-of-the-art for CCG-supertagging and pure dependency parsing (i.e., without using constituency parse annotations) and results competitive with the current state-of-the-art on the other tasks. Our dependency parsing result is particularly important because our model does not include part-of-speech tags as input, which other works have shown to improve performance notably (Dozat & Manning, 2017; Chen & Manning, 2014). Of the prior results listed in the Table 2, only TagLM from Peters et al. (2017) is semi-supervised. However, their approach relies on pre-training rather than self-training: their model incorporates representation produced by an enormous separately-trained language model with 8192 hidden units. Our models use 1024 hidden units in their largest LSTMs, so they are many times faster to run.

| Method | CCG | Chunk | NER | POS |
|---|---|---|---|---|
| C2W + LSTM[a] | – | – | – | **97.78** |
| LSTM-CNN-CRF[b] | – | – | 91.21 | 97.55 |
| Tri-Trained LSTM[c] | 94.7 | – | – | – |
| Shortcut LSTM[d] | 95.08 | – | – | 97.53 |
| JMT[e] | – | 95.77 | – | 97.55 |
| TagLM-2048[f*] | – | – | $91.66 \pm 0.23$ | – |
| TagLM[f] | – | $\mathbf{96.37 \pm 0.05}$ | $\mathbf{91.93 \pm 0.19}$ | – |
| LM-LSTM-CNN-CRF[g] | – | $95.96 \pm 0.08$ | $91.71 \pm 0.10$ | $97.53 \pm 0.03$ |
| Baseline (supervised) | $94.98 \pm 0.05$ | $95.03 \pm 0.02$ | $90.86 \pm 0.03$ | $97.65 \pm 0.02$ |
| Baseline + LM | $95.08 \pm 0.05$ | $95.64 \pm 0.07$ | $91.35 \pm 0.07$ | $97.70 \pm 0.01$ |
| Consistency | $95.05 \pm 0.03$ | $94.82 \pm 0.05$ | $90.84 \pm 0.14$ | $97.61 \pm 0.02$ |
| CVT, no $p_\theta^{\text{future}}$ or $p_\theta^{\text{past}}$ | $95.34 \pm 0.04$ | $95.63 \pm 0.08$ | $91.21 \pm 0.07$ | $97.67 \pm 0.02$ |
| CVT, no $p_\theta^{\text{fwd}}$ or $p_\theta^{\text{bwd}}$ | $\mathbf{95.43 \pm 0.07}$ | $95.88 \pm 0.09$ | $91.62 \pm 0.09$ | $\mathbf{97.76 \pm 0.01}$ |
| CVT | $\mathbf{95.49 \pm 0.04}$ | $96.10 \pm 0.12$ | $91.66 \pm 0.10$ | $\mathbf{97.76 \pm 0.01}$ |
| CVT + LM | $\mathbf{95.49 \pm 0.03}$ | $96.15 \pm 0.09$ | $\mathbf{92.08 \pm 0.10}$ | $\mathbf{97.76 \pm 0.01}$ |

[a]Ling et al. (2015) [b]Ma & Hovy (2016) [c]Lewis et al. (2016) [d]Wu et al. (2017) [e]Hashimoto et al. (2017) [f]Peters et al. (2017) [g]Liu et al. (2017)
*The full TagLM model has many times more parameters than ours. TagLM-2048 is of more comparable size to our models, although still larger.

Table 2: Results for sequence tagging tasks. We report the means and standard deviation of 5 runs. "Baseline" trains with $\mathcal{L}_{\text{sup}}$, "Consistency" trains with" $\mathcal{L}_{\text{sup}} + \mathcal{L}_{\text{consistency}}$, and "CVT" trains with $\mathcal{L}_{\text{sup}} + \mathcal{L}_{\text{CVC}}$. +LM means language modeling is added as an auxiliary task on the unlabeled data.

| Method | Depparse UAS | Depparse LAS |
|---|---|---|
| Hashimoto et al. (2017) | 94.67 | 92.90 |
| Ma & Hovy (2017) | 94.9 | 93.0 |
| Shi et al. (2017) | 95.33 | – |
| Dozat & Manning (2017) | 95.74 | 94.08 |
| Baseline (supervised) | $95.04 \pm 0.04$ | $93.22 \pm 0.05$ |
| Baseline + LM | $95.53 \pm 0.06$ | $93.79 \pm 0.05$ |
| Consistency | $95.31 \pm 0.06$ | $93.39 \pm 0.07$ |
| CVT | $95.73 \pm 0.06$ | $94.00 \pm 0.04$ |
| CVT + LM | $\mathbf{95.99 \pm 0.07}$ | $\mathbf{94.30 \pm 0.06}$ |

Table 3: Results for dependency parsing. We omit results from Choe & Charniak (2016), Kuncoro et al. (2017), and Liu & Zhang (2017) because these train constituency parsers and convert the system outputs to dependency parses. They produce higher scores, but have access to more information during training and do not apply to datasets without constituency annotations.

Although the large TagLM model is competitive with ours for Chunking and NER, reducing the size of TagLM to having 2048 hidden units already causes it to perform worse than our model.

Although there has been a large body of work successfully applying consistency-cost-based learning to vision tasks, we find it does not provide the same gains for NLP. Training a model with the consistency loss $\mathcal{L}_{\text{consistency}}$ did not improve over the baseline for sequence tagging and only slightly improved over the baseline for dependency parsing. This result is perhaps due to the lack of benefit from adding noise when the input consists of discrete tokens as discussed earlier. CVT, on the other hand, works well as a semi-supervised learning method for NLP.

**Importance of Auxiliary Prediction Layers.** To determine which of the auxiliary prediction layers are most valuable for sequence tagging, we do a brief ablation study by training models without the $p_\theta^{\text{fwd}}/p_\theta^{\text{bwd}}$ or $p_\theta^{\text{future}}/p_\theta^{\text{past}}$ auxiliary softmax layers. We find that both kinds of layers improve perfor-

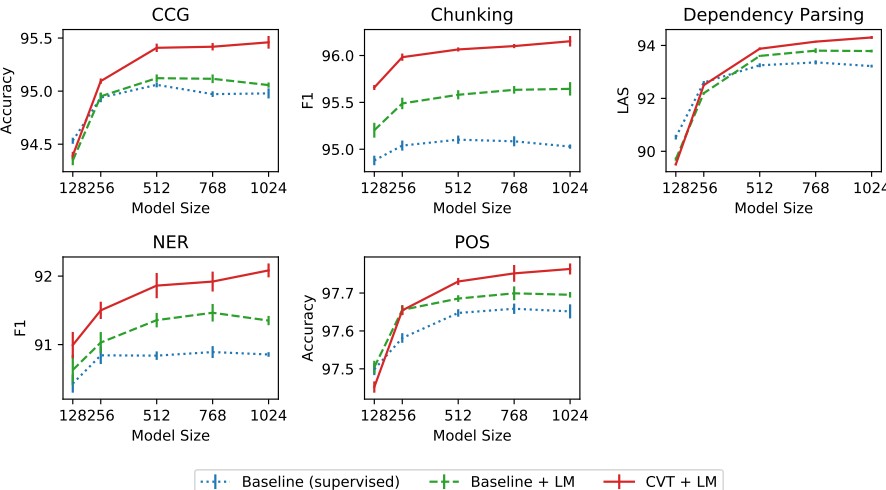

Figure 4: Accuracy vs. size of the LSTMs in the first BiLSTM layer; the ones in the second layer (and MLPs in the case of dependency parsing) are half the size of the ones in the first layer. Points and error bars correspond to means and standard deviations over 5 runs.

mance, but the "future" and "past" layers are more beneficial than the "forward" and "backward" ones, perhaps because theses provide a more distinct and challenging view of the input.

**Training Larger NLP Models.** Most sequence taggers and dependency parsers in prior use work small LSTMs (hidden state sizes of at most 500 units) because larger models yield little to no gains in performance (Reimers & Gurevych, 2017). We found our own supervised approaches and, to a lesser extent, our models when only using language modeling as the auxiliary task to also not benefit from increasing the model size. In contrast, when using CVT accuracy scales much better with model size (see Figure 4). This result suggests the appropriate semi-supervised learning methods may enable the development of larger, more sophisticated models for natural language processing tasks with limited amounts of labeled data.

## 5 CONCLUSION

We propose Cross-View Training, a new method for semi-supervised learning. Our approach allows models to effectively leverage their own predictions on unlabeled data. We report excellent results on semi-supervised image recognition benchmarks and five tasks from natural language processing. We see the development of CVT architectures for other tasks and theoretical analysis of CVT as potential areas of future work.

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

## A    SEQUENCE TAGGING MODEL DETAILS

Our sequence tagging model is a two layer CNN-BiLSTM (Chiu & Nichols, 2016; Ma & Hovy, 2016; Lample et al., 2016). The model produces a representation for each word in the input sentence as follows. First, the characters in the word are embedded, resulting in a sequence of vectors. Then a 1-dimensional convolution over these vectors followed by a max-pooling operation produces a character-level representation of the word. Lastly, this character-based representation is added to a word vector from an embedding matrix. The resulting sequence of word embeddings is then fed through two BiLSTM layers and a softmax layer to produce an output distribution over labels for each token.

We apply dropout (Hinton et al., 2012) to the word embeddings and outputs of each BiLSTM. We apply label smoothing (Szegedy et al., 2016; Pereyra et al., 2017) to the target labels. We use an exponential-moving-average (EMA) of the model weights during training as the final model; we

found this to slightly improve accuracy and significantly reduce the variance in accuracy between models trained with different random initializations. For Chunking and Named Entity Recognition, we use a BIOES tagging scheme. The model is trained using SGD with momentum (Polyak, 1964; Sutskever et al., 2013). Word embeddings are initialized with GloVe vectors (Pennington et al., 2014). During training, we alternate minimizing $\mathcal{L}_{\text{sup}}$ over a minibatch of supervised examples and minimizing $\mathcal{L}_{\text{CVC}}$ over a minibatch of unlabeled examples. The full set of model hyperparameters are listed below.

| Parameter | Value |
|---|---|
| Word Embeddings Initializiation | 300d GloVe[2] |
| Character Embedding Size | 100 |
| Character CNN Filter Widths | [2, 3, 4] |
| Character CNN Num Filters | 300 (100 per filter width) |
| LSTM sizes | 1024 for the first layer, 512 for the second one |
| Language model vocabulary size | 10,000 (only applies to the +LM models) |
| Dropout | 0.5 for labeled examples, 0.8 for unlabeled examples |
| Label Smoothing | 0.1 for labeled examples, none for unlabeled examples |
| EMA coefficient | 0.998 |
| Learning rate | $0.5/(1 + 0.005t^{0.5})$ ($t$ is number of SGD updates so far) |
| Momentum | 0.9 |
| Batch size | 64 |
| Stopping criteria | 150,000 updates |

Table 4: Hyperparemters for NLP models.

## B  DEPENDENCY PARSING MODEL DETAILS

We use the same 2-layer CNN-BiLSTM encoder and the same hyperparameters as listed in Appendix A. The MLPs used to produce representations for candidate head and dependent words have one hidden layer of size 512 with a ReLU activation and an output layer of size 256. We apply dropout to the output of the hidden layer. We omit punctuation from evaluation, which is standard practice for the PTB-SD 3.3.0 dataset.

## C  CROSS-VIEW AUXILIARY LOSSES FOR SUPERVISED LEARNING

In initial experiments, we explored whether cross-view losses could benefit purely supervised classifiers. To do this, we trained models with the following objective:

$$\mathcal{L}_{\text{sup-cv}} = \frac{1}{|\mathcal{D}_l|} \sum_{x_i, y_i \in \mathcal{D}_l} \left( CE(y_i, p_\theta(y|x_i)) + \frac{\lambda_1}{k} \sum_{j=1}^{k} CE(y_i, p_\theta^j(y|x_i)) \right)$$

See Section 3.1 for an explanation of the notation. We hoped that adding auxiliary softmax layers with different views of the input would act as a regularizer on the model. However, we found little to no benefit from this approach. For sequence tagging, results improved slightly on CCG and POS but degraded on NER and Chunking. For image recognition, we augmented WideResNet (Zagoruyko & Komodakis, 2016) with auxiliary softmax layers and evaluated it on CIFAR-10 and CIFAR-100. On both datasets, the augmented model performed slightly worse (by ~0.2% on CIFAR-10 and ~0.9% on CIFAR-100).

We also experimented with using $\mathcal{L}_{\text{sup-cv}}$ instead of of $\mathcal{L}_{\text{sup}}$ on semi-supervised CIFAR-10 and CIFAR-10+. Surprisingly, it (slightly) decreased performance for all of the methods we experimented with: supervised training, VAT, CVT, and CVT with adversarial noise. We note we only tried these experiments with $\lambda_1 = 1$, but this value of $\lambda_1$ did work well for the semi-supervised setting. These negative results suggest that the gains are from CVT are from the improved self-training mechanism, not the additional prediction layers regularizing the model.

