# OpenReview forum: "Cross-View Training for Semi-Supervised Learning"
_ICLR.cc/2018/Conference — Invite to Workshop Track_

### Official Review · AnonReviewer1 · 2017-11-27
**This paper presents a so-called cross-view training for semi-supervised deep models. This paper suffers from several weaknesses, e.g., lack of novelty, technical flaw and no significant improvement over the existing approaches.**

**Rating:** 2
**Confidence:** 4

**Review:**

This paper presents a so-called cross-view training for semi-supervised deep models. Experiments were conducted on various data sets and experimental results were reported.

Pros:
* Studying semi-supervised learning techniques for deep models is of practical significance.

Cons:
* The novelty of this paper is marginal. The use of unlabeled data is in fact a self-training process. Leveraging the sub-regions of the image to improve performance is not new and has been widely-studied in image classification and retrieval.
* The proposed approach suffers from a technical weakness or flaw. For the self-labeled data, the prediction of each view is enforced to be same as the assigned self-labeling. However, since each view related to a sub-region of the image (especially when the model is not so deep), it is less likely for this region to contain the representation of the concepts (e.g., some local region of an image with a horse may exhibit only grass); enforcing the prediction of this view to be the same self-labeled concepts (e.g,“horse”) may drive the prediction away from what it should be ( e..g, it will make the network to predict grass as horse). Such a flaw may affect the final performance of the proposed approach.
* The word “view” in this paper is misleading. The “view” in this paper is corresponding to actually sub-regions in the images
* The experimental results indicate that the proposed approach fails to perform better than the compared baselines in table 2, which reduces the practical significance of the proposed approach.

---

> ### Author Response · Authors · 2017-12-13
> **Response**
>
> Thank you for the comments! We would like to address the cons you listed in order:
>
> 1. “The novelty of this paper is marginal.”:
> To the best of our knowledge the contribution to NLP is completely novel. We actually consider our NLP results to be more important than our image recognition ones because (1) they use external unlabeled data instead artificially making the dataset semi-supervised (2) they are on more widely-used tasks and (3) although the past few years of development on consistency-cost-based and GAN-based semi-supervised learning methods have yielded gains in accuracy for image classification, they are not effective for sequence tagging (whereas our method is).
>
> “Leveraging the sub-regions of the image to improve performance is not new and has been widely-studied in image classification and retrieval.”
> We believe leveraging sub-regions of the image to improve semi-supervised learning is novel, even though leveraging sub-regions has been used in prior works on supervised learning
>
> 2. “The proposed approach suffers from a technical weakness or flaw.”
> The reviewer’s comment on the technical flaw applies to image recognition, but not NLP. We also note even the smallest views in our model see a 21x21 region of the 32x32 images, so it is unlikely for a view to contain no representative concepts. But even aside from these two points we disagree with the criticism. This same “technical flaw” exists (although to a less degree) for any CNN with global mean pooling (an extremely common architecture). Like with our method, a mean-pooled CNN will encourage the feature vectors extracted from all patches of the image to be representative of the target class, not just the ones from the most salient patches. However, we think in many cases this is a good thing rather than a bad one: on difficult examples it is beneficial for the model to leverage the context surrounding the main part of the image (e.g., that an animal is standing in a field of grass) to better classify it (e.g., as a horse rather than a cat).
>
> 3. “The word “view” in this paper is misleading. The “view” in this paper is corresponding to actually sub-regions in the images”
> A view being a sub-region of the image is true in the case of image recognition, but obviously not for NLP. We use “view” as a general term for particular subset of the input features. This usage of “view” is from Blum and Mitchell’s very influential paper “Combining Labeled and Unlabeled Data with Co-Training,” so “view” is terminology that has been around since 1998.
>
> 4. “The experimental results indicate that the proposed approach fails to perform better than the compared baselines in table 2”
> CVT significantly outperforms our baselines. If the reviewer is using “baselines” to refer to prior work, we note (as we mention in the paper) that the TagLM model has far more parameters than ours (LSTMS with up to 8 times as many hidden units) and thus is also many times slower than ours for training and inference. When using a model with only twice as many hidden units as ours, their results drop to significantly below our numbers (see Table 6 in their paper). Therefore we believe their results are close to ours because their models are much larger, not because their method is equally effective.

---

### Official Review · AnonReviewer3 · 2017-11-28
**Experimental results are not impressive**

**Rating:** 5
**Confidence:** 4

**Review:**

The paper proposes a ’Cross View training’ approach to semi-supervised learning. In the teacher-student framework for semi-supervised learning, it introduces a new cross view consistency loss that includes auxiliary softmax layers (linear layers followed by softmax) on lower levels of the student model. The auxiliary softmax layers take different views of the input for prediction.

Pros:
1. A simple approach to encourage better representations learned from unlabeled examples.

2. Experiments are comprehensive.

Cons:

0. The whole paper just presented strategies and empirical results. There are no discussions of insights and why the proposed strategy work, for what cases it will work, and for what cases it will not work? Why?

1. The addition of auxiliary layers improves Sequence Tagging results marginally.

2. The claim of cross-view for sequence tagging setting is problematic. Because the task is per-position tagging, those added signals are essentially not part of the examples, but the signals of its neighbors.

3. Adding n^2 linear layers for image classification essentially makes the model much larger. It is unfair to compare to the baseline models with much fewer parameters.

4. The "CVT, no noise" should be compared to "CVT, random noise", then to "CVT, adversarial noise". The current results show that the improvements are mostly from VAT, instead of CVT.

---

> ### Author Response · Authors · 2017-12-13
> **Response**
>
> Response: Thank you for the comments! We would like to address the cons you listed:
>
> 0. “There are no discussions of insights and why the proposed strategy work”
> We discuss in the abstract and introduction why CVT works. To reiterate, there is a mutually beneficial relationship between the teacher and the students. The students can learn from the teacher because the teacher has access to more of each input and thus produces more accurate labels. Meanwhile, as the students learn they improve the representations for the parts of the input they are exposed to. These better representations in turn improve the teacher. In Section 4.1 under “Model Analysis” we present further insights into why the method works by analyzing the behavior of the trained models.
>
> “...for what cases it will work, and for what cases it will not work”
> We believe CVT will be less effective if the views are too restricted (e.g., seeing very small patches of an input image, in which case the auxiliary prediction layers will not be able to learn effectively) or the views are too unrestricted (e.g., seeing almost the entire image, in which case the auxiliary layers will be very similar to the teacher and thus not be able to benefit from the teacher’s predictions).
>
> 1. “The addition of auxiliary layers improves Sequence Tagging results marginally. “
>  Although in absolute terms the gains are small, performance in sequence tagging is quite saturated, making large gains difficult to achieve. Looking at improvements over baselines in prior work, Wu et al., (2017) report gains of 0.3 for CCG and 0.05 for POS; Liu et al. (2017) report gains of 0.16 for Chunking, 0.49 for NER, and 0.09 for POS; Hashimoto et al. (2017) report gains of 0.75 for Chunking and 0.10 for POS-tagging; Peters et al. (2017), report gains of 1.37 for Chunking and 1.06 for NER. Therefore our gains (comparing “Baseline” vs “CVT” in Table 2) of 0.51 for CCG, 1.07 for Chunking, 0.80 for NER, and 0.11 for POS are pretty large in the context of sequence tagging research. We also note that the large gains from Peters et al. come from using a model many times bigger than ours. When they apply their method to a model more comparable to ours in size, their gains are smaller (see Table 6 of their paper).
>
> 2. “The claim of cross-view for sequence tagging setting is problematic.”
> We are not quite sure what the reviewer means by “problematic.” It is completely normal to leverage a token’s context (i.e., surrounding tokens) when making predictions for sequence tagging. Our “future’’ and “past” auxiliary losses improve this contextual information (which gets passed to the primary softmax layer through the BiLSTMs), resulting in better accuracy.
>
> 3. “It is unfair to compare to the baseline models with much fewer parameters”
> The extra parameters are only used at training-time, so we don’t think it’s an unfair comparison. The models have exactly the same expressive power because they have the same set of test-time parameters. We also note the additional layers only contain about 15% of the model’s parameters for image classification and about 5% for sequence tagging.
>
> 4.  "The "CVT, no noise" should be compared to "CVT, random noise""
> This is a good point, and we will add that comparison!
>
> "The current results show that the improvements are mostly from VAT, instead of CVT."
> Although the improvements for image recognition are larger for VAT than CVT, CVT still works well as a semi-supervised learning method on its own while training almost twice as fast as VAT (which requires two backwards passes for each minibatch instead of just one). We also note that we were unable to get VAT working for sequence tagging, so we believe our method has the advantage of being more applicable to NLP tasks.

---

> > ### Author Response · Authors · 2017-12-27
> > **"CVT, random noise" results**
> >
> > We have updated our paper with "CVT, random noise" results for the vision tasks as the reviewer suggested. CVT with random input noise works almost as well as VAT, suggesting the improvements from CVT are close to the improvements from VAT. However, the additional computation cost for CVT is much smaller than the additional computation cost for VAT.

---

### Official Review · AnonReviewer2 · 2017-11-28
**This paper proposes a multi-view semi-supervised method**

**Rating:** 7
**Confidence:** 4

**Review:**

This paper proposes a multi-view semi-supervised method. For the unlabelled data, a single input (e.g., a picture) is partitioned into k new inputs permitting overlap. Then a new objective is to obtain k predictions as close as possible to the prediction from the model learned from mere labeled data.

To be more precise, as seen from the last formula in section 3.1, the most important factor is the D function (or KL distance used here). As the author said, we could set the noisy parameter in the first part to zero, but have to leave this parameter non-zero in the second term. Otherwise, the model can't learn anything.

My understanding is that the key factor is not the so called k views (as in the first sight, this method resembles conventional ensemble learning very much), but the smoothing distribution around some input x (consistency related loss). In another word, we set the k for unlabeled data as 1, but use unlabeled data k times in the scale (assuming no duplicate unlabeled data), keeping the same training (consistency objective) method, would this new method obtain a similar performance? If my understanding is correct, the authors should further discuss the key novelty compared to the previous work stated in the second paragraph of section 1. One obvious merit is that the unlabeled data is utilized more efficiently, k times better.

---

> ### Author Response · Authors · 2017-12-13
> **Response**
>
> Thank you for the comments!
>
> “...would this new method obtain a similar performance?”
> We think the model definitely does benefit from using more than one view. For example, for sequence tagging adding “forward” and “backward’ views on top of the “future” and “past” views improved performance (see Table 2). We believe you could perhaps set k=1 and get good results if you sampled a different a view for each example, but this would cause the model to train much slower than when learning from all views simultaneously.

---

### Author Response · Authors · 2017-12-13
**General Comments + New SOTA for Graph-Based Dependency Parsing**

We want to emphasize that CVT is applicable to multiple domains, achieving state-of-the-art results for NLP tasks as well as vision ones. We believe our results on NLP tasks are particularly important because:
(1) They use external unlabeled data instead of artificially making the dataset semi-supervised (as in standard semi-supervised vision benchmarks).
(2) Tasks like dependency parsing and NER have been studied for decades and are widely used in industry (whereas CIFAR-10 is a bit more of a "toy" task).
(3) The discrete structure of language makes applying many recent semi-supervised learning methods difficult. Many prior works (e.g., all seven papers in Table 1) only evaluate on vision tasks. As we discuss in our paper, we were unable to successfully apply these methods to NLP tasks successfully.

To further demonstrate the utility of our method, we recently applied CVT to dependency parsing and achieved excellent results. We use a graph-based dependency parser similar to the one from Dozat and Manning (ICLR 2017). CVT improves over a fully supervised system by 0.7 LAS points on the Penn Treebank (using Stanford Dependencies), and achieves a new state-of-the-art for graph-based dependency parsing. We will update the paper with these results soon.

---

### Author Response · Authors · 2017-12-21
**Updated Manuscript**

We have updated our paper to include the new dependency parsing results.

---

### Decision · Program_Chairs · 2018-01-29
**ICLR 2018 Conference Acceptance Decision**

**Decision:**

Invite to Workshop Track

**Comment:**

This paper combines ideas from student-teacher training and multi-view learning in a simple but clever way.  There is not much novelty in the methods, but promising results are given across several tasks, including realistic NLP tasks.  The improvements are not huge but are consistent.  Considering the limited novelty, the paper should include some more convincing analysis and insight on why/when the approach works. Given the intersting results, the committee recommends this for workshop track.